

**Contrasting behaviors of the atmospheric CO₂ interannual**
**variability during two types of El Niños**
Jun Wang[1,2], Ning Zeng[2,3], Meirong Wang[4], Fei Jiang[1], Jingming Chen[1,5], Pierre
Friedlingstein[6], Atul K. Jain[7], Ziqiang Jiang[1], Weimin Ju[1], Sebastian Lienert[8,9], Julia
Nabel[10], Stephen Sitch[11], Nicolas Viovy[12], Hengmao Wang[1], Andrew J. Wiltshire[13]
[1]International Institute for Earth System Science, Nanjing University, Nanjing, China
[2] State Key Laboratory of Numerical Modelling for Atmospheric Sciences and Geophysical Fluid
Dynamics, Institute of Atmospheric Physics, Beijing, China
[3]Department of Atmospheric and Oceanic Science and Earth System Science Interdisciplinary
Center, University of Maryland, College Park, Maryland, USA
[4]Joint Center for Data Assimilation Research and Applications/Key Laboratory of Meteorological
Disaster of Ministry of Education, Nanjing University of Information Science & Technology,
Nanjing, China
[5]Department of Geography, University of Toronto, Ontario M5S3G3, Canada
[6]College of Engineering, Mathematics and Physical Sciences, Unvernity of Exeter, Exeter EX4
4QE, UK
[7]Department of Atmosheric Sciences, University of Illinois at Urbana-Champaign, Urbana, IL
61801, USA
[8]Climate and Environmental Physics, Physics Institute, University of Bern, Bern, Switzerland
[9]Oeschger Centre for Climate Change Research, University of Bern, Bern, Switzerland
[10]Land in the Earth System, Max Planck Institute for Meteorology, D-20146 Hamburg, Germany
[11]College of Life and Environmental Sciences, University of Exeter EX4 4QF, UK
[12]Laboratoire des Sciences du Climat et de l'Environnement, LSCE/IPSL-CEA-CNRS-UVQS,
F-91191, Gif sur Yvette, France
[13]Met office Hadley Centre, Fitzroy Rd, Exeter. EX1 3PB. UK
**Correspondence to: (Ning Zeng, zeng@umd.edu; Fei Jiang, jiangf@nju.edu.cn)**



**Abstract**
El Niño has two different flavors: eastern Pacific (EP) and central Pacific (CP) El
Niños, with different global teleconnections. However, their different impacts on
carbon cycle interannual variability remain unclear. We here compared the behaviors
of the atmospheric $CO_2$ interannual variability and analyzed their terrestrial
mechanisms during these two types of El Niños, based on Mauna Loa (MLO) $CO_2$
growth rate (CGR) and Dynamic Global Vegetation Models (DGVMs) historical
simulations. Composite analysis shows that evolutions of MLO CGR anomaly have
three clear differences in terms of (1) negative and neutral precursors in boreal spring
of El Niño developing years (denoted as "yr0"), (2) strong and weak amplitudes, and
(3) durations of peak from December (yr0) to April of El Niño decaying year (denoted
as "yr1") and from October (yr0) to January (yr1) during EP and CP El Niños,
respectively. Models simulated global land–atmosphere carbon flux ($F_{TA}$) is able to
capture the essentials of these characteristics. We further find that the gross primary
productivity (GPP) over the tropics and extratropical southern hemisphere (Trop+SH)
generally dominates the global $F_{TA}$ variations during both El Niño types. Regionally,
significant anomalous carbon uptake caused by more precipitation and colder
temperature, corresponding to the negative precursor, occurs between 30°S and 20°N
from January (yr0) to June (yr0), while the strongest anomalous carbon releases, due
largely to the reduced GPP induced by low precipitation and warm temperature,
happen between equator and 20°N from February (yr1) to August (yr1) during EP El
Niño events. In contrast, during CP El Niño events, clear carbon releases exist
between 10°N and 20°S from September (yr0) to September (yr1), resulted from the
widespread dry and warm climate conditions. Different spatial patterns of land
temperature and precipitation in different seasons associated with EP and CP El Niños



account for the characteristics in evolutions of GPP, terrestrial ecosystem respiration
(TER), and resultant $F_{TA}$. Understanding these different behaviors of the atmospheric
$CO_2$ interannual variability along with their terrestrial mechanisms during EP and CP
El Niños is important because CP El Niño occurrence rate might increase under
global warming.

**1    Introduction**
The El Niño–Southern Oscillation (ENSO), a dominant year-to-year climate
variability, leads to a significant interannual variability in the atmospheric $CO_2$
growth rate (CGR) (Bacastow, 1976;Keeling et al., 1995). Many studies, including
measurement campaigns (Lee et al., 1998;Feely et al., 2002), atmospheric inversions
(Bousquet et al., 2000;Peylin et al., 2013), and terrestrial carbon cycle models (Zeng
et al., 2005;Wang et al., 2016), consistently suggested the dominant role of terrestrial
ecosystems, especially of tropical ecosystems, to the atmospheric $CO_2$ interannual
variability. Recently, Ahlstrom et al. (2015) further suggested ecosystems over the
semi-arid regions played the most important role in the interannual variability of the
land $CO_2$ sink. Moreover, this ENSO-related carbon cycle interannual variability may
be enhanced under global warming, with an about 44% increase in the sensitivity of
terrestrial carbon flux to ENSO (Kim et al., 2017).
Tropical climatic variations (especially in surface air temperature and precipitation)
induced by ENSO and responses of plant/soil physiology can largely account for the
terrestrial carbon cycle interannual variability (Zeng et al., 2005;Wang et al.,
2016;Jung et al., 2017). Multi-model simulations involved in the TRENDY project
and the Coupled Model Intercomparison Project Phase 5 (CMIP5) have consistently
suggested the biological dominance of the gross primary productivity (GPP) or net





primary productivity (NPP) (Kim et al., 2016;Wang et al., 2016;Piao et al.,
2013;Ahlstrom et al., 2015). However, debates have continued about which is the
dominant climatic mechanism (temperature or precipitation) in the interannual
variability of the terrestrial carbon cycle (Wang et al., 2013;Wang et al., 2014;Cox et
al., 2013;Zeng et al., 2005;Ahlstrom et al., 2015;Wang et al., 2016;Qian et al.,
2008;Jung et al., 2017).
The atmospheric CGR or land–atmosphere carbon flux ($F_{TA}$ – positive sign meaning a
flux into the atmosphere) can anomalously increase during El Niño, and decrease
during La Niña episodes (Zeng et al., 2005;Keeling et al., 1995). Cross correlation
analysis shows that the atmospheric CGR and $F_{TA}$ lags the ENSO by several months
(Qian et al., 2008;Wang et al., 2013;Wang et al., 2016), because of the period needed
for surface energy and soil moisture adjustment following ENSO-related circulation
and precipitation anomalies (Gu and Adler, 2011;Qian et al., 2008). However,
considering the ENSO diversity (Capotondi et al., 2015), the atmospheric CGR and
$F_{TA}$ can show different behaviors during different El Niño events (Schwalm,
2011;Wang et al., 2018).
In climate, El Niño events can be classified into eastern Pacific El Niño (EP El Niño,
also termed as conventional El Niño) and central Pacific El Niño (CP El Niño, also
termed as El Niño Modoki), according to the patterns of sea-surface warming over the
tropical Pacific (Ashok et al., 2007;Ashok and Yamagata, 2009). These two types of
El Niño have different global climatic teleconnections, associated with contrasting
climate conditions in different seasons (Weng et al., 2007;Weng et al., 2009). For
example, positive winter temperature anomalies are located mostly over the
northeastern US during EP El Niño, while warm anomalies are in northwestern US
during CP El Niño (Yu et al., 2012). The contrasting summer and winter precipitation



anomaly patterns associated with these two El Niño events over the China, Japan, and
US were also presented by Weng et al. (2007; 2009). Importantly, Ashok et al. (2007)
suggested that the occurrence of CP El Niño had increased during recent decades, as
compared to EP El Niño. This phenomenon can probably be attributed to the
anthropogenic global warming (Ashok and Yamagata, 2009;Yeh et al., 2009).
However, the contrasting impacts of EP and CP El Niño events on the carbon cycle
variability remain unclear. In this study, we attempt to reveal their different impacts.
Therefore, we carefully compared the behaviors of the atmospheric $CO_2$ interannual
variability and analyzed their terrestrial mechanisms corresponding to these two types
of El Niños, based on Mauna Loa long-term CGR and TRENDY multi-model
simulations.
This paper is organized as follows: Section 2 describes the datasets used, methods,
and TRENDY models selected. Section 3 show the results about the relationship
between ENSO and CGR, EP and CP El Niño events, composite analysis on carbon
cycle behaviors, and terrestrial mechanisms. Some discussions will be presented in
Section 4, and concluding remarks are in Section 5.

**2 Datasets and Methods**
**2.1 Datasets used**
We accessed the monthly atmospheric $CO_2$ concentration between 1960 and 2013
from the National Oceanic and Atmospheric Administration (NOAA) Earth System
Research Laboratory (ESRL). The annual $CO_2$ growth rate (CGR) in Pg C $yr^{-1}$ is
derived month by month according to the approach (Patra et al., 2005;Sarmiento et al.,

127   2010)



$$CGR(t) = \gamma \cdot [pCO_2(t+6) - pCO_2(t-6)] \qquad (1)$$

where $\gamma = 2.1276$ Pg C ppm$^{-1}$, $pCO_2$ is the atmospheric partial pressure of $CO_2$ in
ppm, t represents the time in months. The detailed calculation of the conversion factor
($\gamma$) can be referred to the appendix (Sarmiento et al., 2010).
We obtained the temperature and precipitation datasets between 1960 and 2013 from
CRUNCEPv6 (Wei et al., 2014). CRUNCEP datasets are the merged product of the
ground observation-based CRU data and model-based NCEP-NCAR Reanalysis data,
with a 0.5°×0.5° spatial and 6 hourly temporal resolution. These datasets are
consistent with the climatic forcing used to run dynamic global vegetation models in
TRENDY v4 (Sitch et al., 2015). The sea surface temperature anomalies (SSTA) over
the Niño3.4 region (5°S–5°N, 120°–170°W) were from the NOAA's Extended
Reconstructed Sea Surface Temperature (ERSST) dataset, version 4 (Huang et al.,

140    2015).

We also took the inversion of $F_{TA}$ from the Jena CarboScope as a comparison with the
TRENDY multi-model simulations from 1981 to 2013. The Jena CarboScope Project
provides the estimates of the surface-atmosphere carbon flux based on the
atmospheric measurements through an "atmospheric transport inversion". The
inversion run used here is the s81_v3.8 (Rodenbeck et al., 2003).

**2.2 TRENDY simulations**
We analyzed eight state-of-the-art dynamic global vegetation models from TRENDY





v4 for the period 1960–2013: CLM4.5 (Oleson et al., 2013), ISAM (Jain et al., 2013),
JSBACH (Reick et al., 2013), JULES (Clark et al., 2011), LPX-Bern (Keller et al.,
2017), OCN (Zaehle and Friend, 2010), VEGAS (Zeng et al., 2005), and VISIT (Kato
et al., 2013) (Table 1). Since LPX-Bern was excluded in the analysis of TRENDY v4,
due to it not fulfilling the minimum performance requirement, the output over the
same time period of a more recent version (LPX-Bern v1.3) was used. These models
were forced by a common set of climatic datasets (CRUNCEPv6) and followed the
same experimental protocol. The 'S3' run was used in this study, in which simulations
forced by all the drivers including the $CO_2$, climate, and land use and land cover
change (Sitch et al., 2015).
We interpolated the simulated terrestrial variables (NBP, GPP, TER, soil moisture etc.)
into a consistent 0.5°×0.5° resolution using the first order conservative remapping
scheme (Jones, 1999) by Climate Data Operators (CDO):
$$\overline{F_k} = \frac{1}{A_k} \int f \, dA \tag{2}$$

where $\overline{F_k}$ denotes the area-averaged destination quantity, $A_k$ is the area of cell $k$, $f$
is the quantity in an old grid which has overlapping area with the destination grid.
Then the median, 5%, and 95% percentiles of multi-model simulations were
calculated grid by grid to study the different effects of EP and CP El Niños on
terrestrial carbon cycle interannual variability.






### 2.3 El Niño criterion and classification methods


El Niño events are determined by the Oceanic Niño Index (ONI) [i.e. the running
3-month mean SST anomaly over the Niño3.4 region]. This NOAA criterion is that El
Niño events are defined as 5 consecutive overlapping 3-month periods at or above the
+0.5° anomaly.
We classified El Niño events into EP or CP based on the consensus of three different
identification methods directly adopted from previous study (Yu et al., 2012). These
identification methods include El Niño Modoki Index (EMI) (Ashok et al., 2007), the
EP/CP-index method (Kao and Yu, 2009), and the Niño method (Yeh et al., 2009).

### 2.4 Anomaly calculation and composite analysis


To calculate the anomalies, we first removed the long-term climatology of the period
1960–2013 from all of the variables, in order to get rid of seasonal cycle signals. We
then detrended them based on a linear regression, because (1) the trend in terrestrial
carbon variables was mainly caused by long-term $CO_2$ fertilization and climate
change, (2) the trend in CGR resulted mainly from the anthropogenic emissions. We
used these detrended monthly anomalies to investigate the impacts of El Niño events
on carbon cycle interannual varibility.
Specifically, we adopted the composite analysis, which is widely used in the climate
research, to compare the behaviors of the carbon flux (CGR, $F_{TA}$ i.e.) based on the
selected EP and CP El Niño events. We use the Bootstrap Methods (Mudelsee, 2010)
to estimate the 95% confidence intervals and the Student's $t$-test to estimate the
significance levels in the composite analysis. The 80% significance level is selected



as used in Weng et al. (2007) due to the limited EP El Niño events.

**3 Results**
**3.1 Relationship between ENSO and atmospheric $CO_2$ interannual variability**
The atmospheric $CO_2$ interannual variability closely couples with ENSO (Fig. 1), with
noticeable increases during El Niño and decreases during La Niña, respectively
(Bacastow, 1976;Keeling and Revelle, 1985). The correlation coefficient between
MLO CGR and Niño3.4 Index from 1960 to 2013 is 0.43 ($p < 0.01$). Regression
analysis further indicates that per unit increase in Nino3.4 Index can lead to 0.60 Pg C
$yr^{-1}$ increase in MLO CGR.
The variation in global $F_{TA}$ anomaly simulated by TRENDY models resembles the
MLO CGR variation, with a correlation coefficient of 0.54 ($p < 0.01$; Fig. 1b). It is
close to the correlation coefficient of 0.61 ($p < 0.01$; Fig. 1b) between MLO CGR
and Jena CarboScope s81 in the periods of 1981–2013. This indicates that the
terrestrial carbon cycle can largely explain the atmospheric $CO_2$ interannual
variability, as suggested by previous studies (Bousquet et al., 2000;Zeng et al.,
2005;Peylin et al., 2013;Wang et al., 2016). Moreover, the correlation coefficient of
TRENDY global $F_{TA}$ and Nino3.4 Index reaches 0.49 ($p < 0.01$) and a similar
regression analysis as done with the MLO CGR shows a sensitivity of 0.64 Pg C $yr^{-1}$
$K^{-1}$. However, owing to the diffuse light fertilization effect induced by the eruption of
Mount Pinatubo in 1991 (Mercado et al., 2009), Jena CarboScope s81 indicates that
the terrestrial ecosystems have an anomalous uptake during 1991/92 El Niño event,



making MLO CGR an anomalous decrease. However, TRENDY models cannot
capture this phenomenon. It is not only due to a lack of a corresponding process
representation in some models, but also because TRENDY protocol does not include
diffuse and direct light forcing.

**3.2 EP and CP El Niño events**
Schematic diagrams of the two types of El Niños (EP and CP) are shown in Fig. 2.
During EP El Niño events (Fig. 2a), a positive sea surface temperature anomaly
(SSTA) occurs in the eastern equatorial Pacific Ocean, showing a dipole SSTA pattern
with the positive zonal SST gradient. This condition forms a single cell of Walker
circulation over the tropical Pacific, with the dry downdraft in the western Pacific and
wet updraft in the central-eastern Pacific. In contrast, the anomalous warming in the
central Pacific, sandwiched by anomalous cooling in the east and west, is observed
during CP El Niño events (Fig. 2b). This tripole SSTA pattern makes the
positive/negative zonal SST gradient in the western/eastern tropical Pacific, resulting
in an anomalous two-cell Walker circulation over the tropical Pacific. This alteration
in atmospheric circulation produces a wet region in the central Pacific. Moreover,
apart from these differences in the equatorial Pacific, the SSTA in other oceanic
regions also differ remarkably (Weng et al., 2007;Weng et al., 2009).
Based on the NOAA criterion, we can detect a total of 17 El Niño events from 1960
till 2013. We then categorize these events into EP or CP El Niño, relying on the
consensus of three identification methods (EMI, EP/CP-index, and Niño methods)





(Yu et al., 2012). Considering the effect of diffuse radiation fertilization induced by
volcano eruptions (Mercado et al., 2009), we remove the 1963/64, 1982/83, and
1991/92 El Niño events, in which Mount Agung, El Chichón, and Pinatubo erupted,
respectively. Further, we closely examined those extended El Niño events (1968/70,
1976/78, 1986/88). Based on the typical responses of MLO CGR to El Niño events
(anomalous increase lasting from the El Niño developing year to El Niño decaying
year; Supplementary Fig. S1), we retained 1968/69, 1976/77, and 1987/88 El Niño
periods. Finally, we got 4 EP El Niño and 7 CP El Niño events in this study (Table 2;
Fig. 1b), with the composite SSTA evolutions in Supplementary Fig. S2.

**3.3 Responses of atmospheric CGR to two types of El Niños**
Based on the selected EP and CP El Niño events, we make the composite analysis
with the non-smoothed detrended monthly anomalies of MLO CGR and TRENDY
global $F_{TA}$ to reveal the contrasting carbon cycle responses to these two types of El
Niños (Fig. 3). Besides the differences in the location of anomalous SST warming
along with the alteration of the atmospheric circulation in EP and CP El Niños shown
in Fig. 2, we find that (1) different El Niño precursors: the SSTA is significant
negative in EP El Niño during the boreal winter (JF) and spring (MAM) in yr0 (yr0
and yr1 refer to the El Niño developing and decaying year, respectively, hereafter),
whereas the SSTA is neutral in CP El Niño; (2) different tendencies of SST
($\partial SST/\partial t$): the tendency of SST in EP El Niño is stronger than that in CP El Niño; (3)
different El Niño amplitudes: due to their different tendencies of SST, the amplitude



of EP El Niño is basically stronger than that of CP El Niño, though they all reach
maturity in November or December in yr0 (Figs. 3a and c).
Correspondingly, behaviors of MLO CGR during these two types of El Niño events
also show some differences (Figs. 3b and d). During EP El Niño events (Fig. 3b), the
MLO CGR is negative in boreal spring (yr0), and increases quickly from boreal fall
(yr0), whereas it is neutral in boreal spring (yr0), and slowly increases from boreal
summer (yr0) during CP El Niño episode (Fig. 3d). The amplitude of the MLO CGR
anomaly during EP El Niño events is generally larger than that during CP El Niño
events; importantly, the duration of MLO CGR peak during EP El Niño is from
December (yr0) to April (yr1), while the MLO CGR anomaly peaks from October
(yr0) to January (yr1) during CP El Niño. Positive MLO CGR anomaly ends around
September (yr1) during both cases (Figs. 3b and d). While finalizing our paper, we
noted the publication of Chylek et al. (2018) who also finds CGR amplitude
difference in response to the two types of events.
Comparing the MLO CGR with the TRENDY global $F_{TA}$ anomalies (Figs. 3b and d),
we can find that TRENDY global $F_{TA}$ can well capture the characteristics of CGR
evolution during the CP El Niño. In contrast, the amplitude of the TRENDY global
$F_{TA}$ anomaly is somewhat underestimated during the EP El Niño, causing lower
significance in statistics (Fig. 3b). This underestimation of global $F_{TA}$ anomaly can,
for example, be clearly seen through the comparison between the TRENDY and Jena
CarboScope during the extreme 1997/98 EP El Niño (Fig. 1b). But the other



characteristics can be captured. Therefore, insight into the mechanisms of these CGR
evolutions during EP and CP El Niños, based on the simulations by TRENDY models,
is still possible.

**3.4 Regional contributions, characteristics, and their mechanisms**
We separate the TRENDY global $F_{TA}$ anomaly by major geographic regions into two
parts: the extratropical northern hemisphere (NH, 23°N–90°N), and tropics plus
extratropical southern hemisphere (Trop+SH, 60°S–23°N) (Fig. 4). Comparing the
contributions of these two parts, we find that the $F_{TA}$ over Trop+SH plays a more
important role in global $F_{TA}$ anomaly during both cases (Figs. 4b and d), consistent
with previous studies (Bousquet et al., 2000;Peylin et al., 2013;Zeng et al.,
2005;Wang et al., 2016;Ahlstrom et al., 2015;Jung et al., 2017). The $F_{TA}$ over
Trop+SH is negative in austral fall (MAM; yr0), increases from austral spring (SON;
yr0), and peaks from December (yr0) to April (yr1) during EP El Niño (Fig. 4b),
whereas it is nearly neutral in austral fall (yr0), increases from austral winter (JJA;
yr0), and peaks from November (yr0) to March (yr1) during CP El Niño (Fig. 4d).
These characteristics of evolutions in $F_{TA}$ over Trop+SH are generally consistent with
the global $F_{TA}$ and MLO CGR (Figs. 3b and d). In contrast, the contributions from the
$F_{TA}$ anomaly over the NH are relatively weaker (or nearly neutral) (Figs. 4a and c).
According to the equation $F_{TA} = -NBP = TER - GPP + D$ (the term D represents
the carbon flux caused by the disturbances such as the wildfires, harvests, grazing,
land cover change etc.), the variation of $F_{TA}$ can be explained by the variations of GPP,



TER, and D. The D simulated by TRENDY is nearly neutral during both El Niño
types (Fig. 4). So GPP and TER can largely account for the variation of $F_{TA}$.
Specifically, in Trop+SH, GPP anomalies dominate the variations of $F_{TA}$ in both El
Niño types, but their evolutions differ (Figs. 4b and d). GPP anomalously increases
during austral fall (yr0), and decreases from austral summer (yr1) to winter (yr1), with
the minimum around April (yr1) during the EP El Niño (Fig. 4b), whereas GPP
anomaly is always negative with the minimum around October or November (yr0)
during the CP El Niño (Fig. 4d). The variation of TER in both El Niños is relatively
weaker than that of GPP (Figs. 4b and d). The anomalous increase during austral
spring (yr0) and summer (yr1) accounts for the increase in $F_{TA}$, and it partly cancels
the decease of GPP in austral fall (yr1) and winter (yr1) during EP El Niño (Fig. 4b).
In contrast, TER has a reduction in yr0 during CP El Niño (Fig. 4d). Over the NH,
though $F_{TA}$ anomaly is relatively weaker, the behaviors of GPP and TER differ in EP
and CP El Niños. GPP and TER consistently decrease in the growing season of yr0
and increase in the growing season of yr1 during EP El Niño (Fig. 4a), whereas they
only show some increase during boreal summer (yr1) during CP El Niño (Fig. 4c).
These evolution characteristics of GPP, TER, and resultant $F_{TA}$ principally result from
their responses to the climate variability. We present the standardized observed
surface air temperature, precipitation, and TRENDY simulated soil moisture content
in Fig.5. Over the Trop+SH, considering the regulation of thermodynamics and
hydrological cycle on surface energy balance, variations of temperature and



precipitation (soil moisture) are always opposite during the two types of El Niños
(Figs. 5b and d). And adjustments of soil moisture lag precipitation for about 2–4
months, owing to the so-called 'soil memory' of water recharge (Qian et al., 2008).
The variations of GPP in both El Niño types are closely associated with variations of
soil moisture, namely water availability largely dominated by precipitation (Figs. 4b
and d, and Figs. 5b and d), consistent with previous studies (Zeng et al., 2005;Zhang
et al., 2016). Warm temperature during El Niño episodes can enhance the ecosystem
respiration, but dry conditions can reduce it. These cancellations from warm and dry
conditions make the amplitude of TER variation smaller than that of GPP (Figs. 4b
and d). Over the NH, variations of temperature and precipitation are basically in the
same direction (Figs. 5a and c), as opposed to their behaviors over the Trop+SH,
because of their different climatic dynamics (Zeng et al., 2005). During the EP El
Niño event, cool and dry conditions in the boreal summer (yr0) inhibit GPP and TER,
whereas warm and wet conditions in the boreal spring and summer (yr1) enhance
them (Fig. 5a, and Fig. 4a). In contrast, only the warm and wet condition in boreal
summer (yr1) enhance GPP and TER during the CP El Niño event. (Fig. 5c and Fig.
4c). These different configurations of temperature and precipitation variations during
EP and CP El Niños form the different evolution characteristics of GPP, TER, and
resultant $F_{TA}$.
Detailed regional evolution characteristics can be seen from the hovmöller diagrams
in Fig. 6 and Supplementary Figs. S3 and S4. The obvious large anomalies of $F_{TA}$



consistently occur from 20°N to 40°S during EP and CP El Niños (Figs. 6c and f),
consistent with above analyses (Figs. 4b and d). Moreover, we can find that there is
clear anomalous carbon uptake between 30°S and 20°N in the periods from January
(yr0) to June (yr0) during the EP El Niño (Fig. 6c), corresponding to the negative
precursor (Fig. 3b and Fig. 4b). This anomalous carbon uptake comparably comes
from the three continents (Supplementary Figs. S4 a–c). Biological process analyses
indicate that GPP dominates between 5°N and 20°N, and between 30°S and 15°S
(Supplementary Fig. S3a), which is related to the more precipitation (Fig. 6b), while
TER dominates between 15°S and 5°N (Supplementary Fig. S3b), largely due to the
colder temperature (Fig. 6a). On the other hand, the strongest anomalous carbon
releases occur between equator and 20°N in the periods from February (yr1) to
August (yr1) during EP El Niño (Fig. 6c). The largest contribution to these anomalous
carbon releases comes from the South America (Supplementary Fig. S4c), and GPP is
the dominant factor to $F_{TA}$ anomaly here (Supplementary Figs. S3a and b). Low
precipitation (with a few months delayed dry conditions; Fig. 6b) and warm
temperature (Fig. 6a) inhibit GPP, causing the positive $F_{TA}$ anomaly (Fig. 6c). In
contrast, the obvious carbon releases can be found between 10°N and 20°S from
September (yr0) to September (yr1) during CP El Niño (Fig. 6f). More specifically,
these clear carbon releases largely come from South America and tropical Asia
(Supplementary Figs. S4 d–f). TER dominates between 15°S and 10°N in the periods
from January (yr1) to September (yr1), and others are dominated by GPP





(Supplementary Figs. S3c and d). Widespread dry and warm conditions (Figs. 6d and
e) can well explain these GPP and TER anomalies, as well as the resultant $F_{TA}$
behavior. For more detailed information on the other regions refer to Supplementary
Figs. S3 and S4.

**4 Discussion**
The El Niño shows large diversity in individual events (Capotondi et al., 2015),
creating the large uncertainties in composite analyses (Figs. 3–5). Moreover, we only
selected four EP El Niño events during the past five decades in this study, which can
be used to research on the carbon cycle interannual variability (Table 1). Owing to the
small samples and large inter-event spread, the statistical significance in the
composite analyses will need to be further evaluated with upcoming EP El Niño
events occurring in the future. However, cross-correlation analyses between long-term
CGR (or $F_{TA}$) and Niño Index have shown that the responses of CGR (or $F_{TA}$) lag
ENSO for a few months (Zeng et al., 2005;Wang et al., 2016;Wang et al., 2013). This
phenomenon can be clearly detected in the EP El Niño composite (Fig. 3b). Therefore,
composite analyses in this study can still give us some insights into the interannual
variability of the global carbon cycle.
Another caveat is that TRENDY models seem to underestimate the amplitude of $F_{TA}$
anomaly during the extreme EP El Niño events (Fig. 1b). This underestimation of $F_{TA}$
may partly result from the bias in estimation of carbon releases induced by wildfires.
As expected, the carbon releases induced by wildfires in strong El Niño events (i.e.



1997/98) played an important role in global carbon variations (van der Werf et al.,
2004;Chen et al., 2017) (Supplementary Fig. S5). But, some TRENDY models (ISAM,
JULES, and OCN) do not include a fire module to explicitly simulate the carbon
releases induced by wildfires (Table 1), and those TRENDY models that contain a fire
module generally underestimate the effects of wildfires. For instance, VISIT and
JSBACH clearly underestimated the carbon flux anomaly induced by wildfires during
the 1997/98 EP El Niño event (Supplementary Fig. S5).
We do not include the recent extreme 2015/16 El Niño event in this study, because the
TRENDY v4 datasets cover the time span from 1860 to 2014. As shown in Wang et al.
(2018), the behavior of MLO CGR in 2015/16 El Niño resembles the composite result
of CP El Niño events (Fig. 3d). But the 2015/16 El Niño event had the extreme
positive SSTA both over the central and eastern Pacific. Its equatorial eastern Pacific
SSTA exceeded +2.0 K, comparable to the historical extreme El Niño events (e.g.
1982/83, 1997/98); the central Pacific SSTA marked the warmest event since the
modern                                   observation
(https://reliefweb.int/report/world/enhancing-resilience-extreme-climate-events-lesson
s-2015-2016-el-ni-o-event-asia-and). Therefore, the 2015/16 El Niño event evolved
not only following the EP El Niño dynamics that relied on the basin-wide thermocline
variations, but also following the CP El Niño dynamics that relied on the subtropical
forcing (Paek et al., 2017;Palmeiro et al., 2017). The 2015/16 extreme El Niño event
can be treated as the strongest mixed EP and CP El Niño, which caused different



climate anomalies compared with the extreme 1997/98 El Niño (Paek et al.,
2017;Palmeiro et al., 2017) with the contrasting terrestrial and oceanic carbon cycle
responses (Wang et al., 2018;Liu et al., 2017;Chatterjee et al., 2017).
Some studies (Yeh et al., 2009;Ashok and Yamagata, 2009) have suggested that CP El
Niño has become or will be more frequent under global warming, compared with EP
El Niño. This shift of El Niño types will alter the response patterns of terrestrial
carbon cycle interannual variability, and encourage us to have further studies in the
future.

**5 Concluding Remarks**
In this study, we investigate the different impacts of EP and CP El Niño events on the
carbon cycle interannual variability in terms of the composite analysis, based on the
long-term MLO CGR and TRENDY multi-model simulations. We suggest that there
are three clear differences in evolutions of MLO CGR during EP and CP El Niños in
terms of their precursor, amplitude, and duration of peak. Specifically, MLO CGR
anomaly is negative in boreal spring (yr0) during EP El Niño events, while it is
neutral during CP El Niño events; the amplitude of the CGR anomaly is generally
larger during EP El Niño events than during CP El Niño events; the duration of MLO
CGR peak during EP El Niño events is about from December (yr0) to April (yr1),
while it peaks from October (yr0) to January (yr1) during CP El Niño events.
TRENDY multi-model simulated global $F_{TA}$ anomalies can basically capture these



characteristics. Further analysis indicates that the $F_{TA}$ anomalies over the Trop+SH
make the most contribution to the global $F_{TA}$ anomalies during these two types of El
Niño events, in which GPP anomalies generally dominate the evolutions of the $F_{TA}$
anomalies rather than TER. Regionally, during EP El Niño events, clear anomalous
carbon uptake occurs between 30°S and 20°N in the periods from January (yr0) to
June (yr0), corresponding to the negative precursor, which is mainly caused by more
precipitation and colder temperature. The strongest anomalous carbon releases happen
between equator and 20°N in the periods from February (yr1) to August (yr1), due
largely to the reduced GPP induced by low precipitation and warm temperature. In
contrast, clear carbon releases exist between 10°N and 20°S from September (yr0) to
September (yr1) during CP El Niño events, which are caused by the widespread dry
and warm climate conditions.

**Data availability.** The monthly atmospheric $CO_2$ concentration is from NOAA/ESRL
(https://www.esrl.noaa.gov/gmd/ccgg/trends/index.html). The Niño3.4 Index is from
ERSST4      (http://www.cpc.ncep.noaa.gov/data/indices/ersst4.nino.mth.81-10.ascii).
Temperature     and     precipitation     are     from     CRUNCEP     v6
(ftp://nacp.ornl.gov/synthesis/2009/frescati/temp/land_use_change/original/readme.ht
m). TRENDY v4 data are available from S. Sitch (s.a.sitch@exeter.ac.uk) upon your
reasonable request.



**Acknowledgements.** We gratefully acknowledge the TRENDY DGVM community,
as part of the Global Carbon Project, for access to gridded land data and the NOAA
ESRL for the use of Mauna Loa atmospheric $CO_2$ records. This study was supported
by the National Key R&D Program of China (grant no. 2017YFB0504000 and no.
2016YFA0600204), the Natural Science Foundation for Young Scientists of Jiangsu
Province, China (Grant No. BK20160625), and the National Natural Science
Foundation of China (Grant No. 41605039). Andrew Wiltshire was supported by the
Joint UK BEIS/Defra Met Office Hadley Centre Climate Programme (GA01101).

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



# Tables and Figures


Table 1 TRENDY models used in this study.

| No. | models | Resolution (lat×lon) | Fire simulation | references |
|-----|--------|---------------------|-----------------|------------|
| 1 | CLM4.5 | 0.94°×1.25° | yes | Oleson et al., 2013 |
| 2 | ISAM | 0.5°×0.5° | no | Jain et al., 2013 |
| 3 | JSBACH | 1.875°×1.875° | yes | Reick et al., 2013 |
| 4 | JULES | 1.6°×1.875° | no | Clark et al., 2011 |
| 5 | LPX-Bern | 1°×1° | yes | Keller et al., 2017 |
| 6 | OCN | 0.5°×0.5° | no | Zaehle et al., 2010 |
| 7 | VEGAS | 0.5°×0.5° | yes | Zeng et al., 2005 |
| 8 | VISIT | 0.5°×0.5° | yes | Kato et al., 2013 |


Table 2 Eastern Pacific (EP) and Central Pacific (CP) El Niño events used in this
study, as identified by the majority consensus of three methods.

| EP El Niño | CP El Niño |
|------------|------------|
| 1972/73 | 1965/66 |
| 1976/77 | 1968/69 |
| 1997/98 | 1987/88 |
| 2006/07 | 1994/95 |
| | 2002/03 |
| | 2004/05 |
| | 2009/10 |






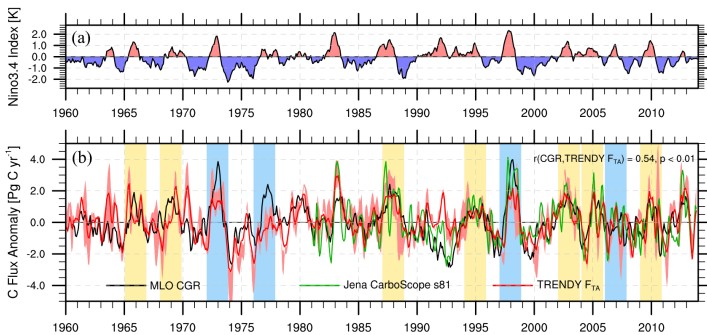


Figure 1. Interannual variability in Niño3.4 Index and carbon cycle. (a) Niño3.4. (b)

Mauna Loa (MLO) $CO_2$ growth rate (CGR, black line), as well as TRENDY

multi-model median (red line) and Jena inversion (green line) of global land–

atmosphere carbon flux ($F_{TA}$, positive value means into the atmosphere, units in Pg C

$yr^{-1}$), which are further smoothed by the 3-month running average. The light red

shaded represents the area between the 5% and 95% percentiles of the TRENDY

simulations. The bars represent the El Niño events selected in this study, with the EP

El Niño in blue and CP El Niño in yellow.


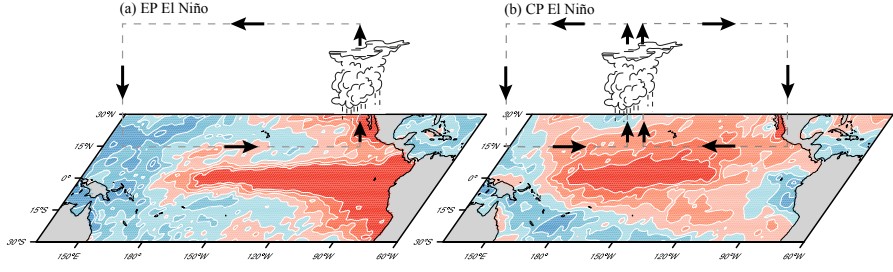


Figure 2. Schematic diagram of two types of El Niños. (a) sea surface temperature

anomaly (SSTA) over the tropical Pacific associated with the anomalous Walker





Circulation in EP El Niño. (b) SSTA with two cells of the anomalous Walker
Circulation in CP El Niño. Red colors indicate warming, and blue colors cooling.
Vectors denote the wind directions.

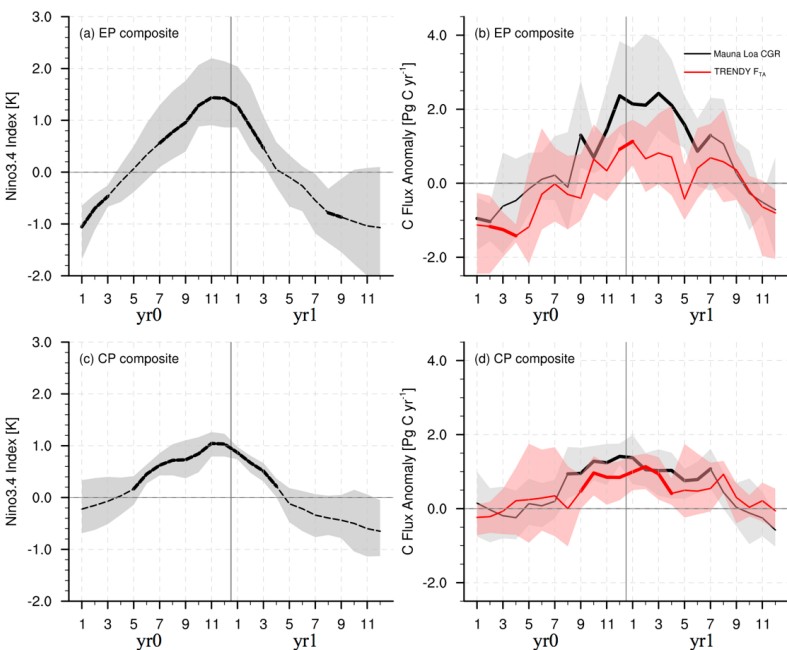


Figure 3. Composites of El Niño and corresponding carbon flux anomaly (Pg C yr$^{-1}$).
(a) Nino3.4 Index composite during EP El Niño events. (b) corresponding MLO CGR
and TRENDY v4 global $F_{TA}$ composite during EP El Niño events. (c) Nino3.4 Index
composite during CP El Niño events. (d) corresponding MLO CGR and TRENDY v4
global $F_{TA}$ composite during CP El Niño events. Shaded area denotes the 95%
confidence intervals of the variables in the composite, derived in 1000 bootstrap
estimates. Bold Lines indicates the significance above the 80% level estimated by



Student's *t*-test.

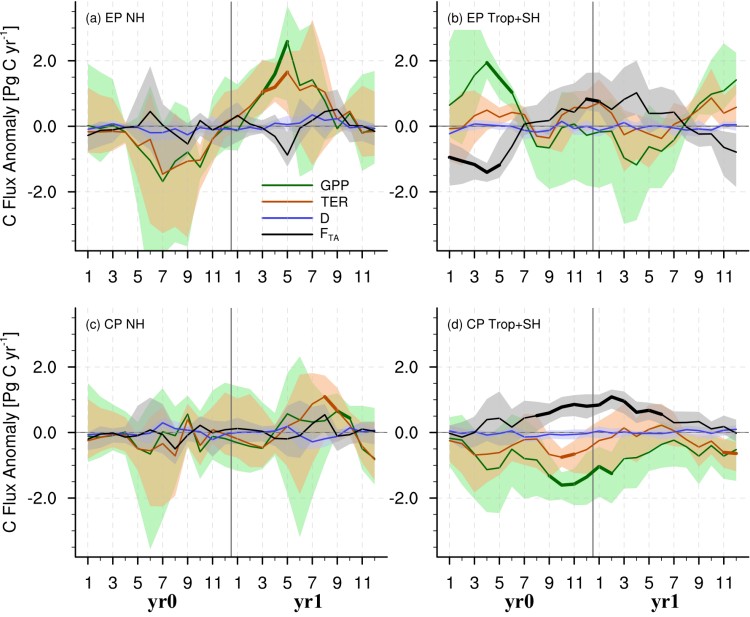


Figure 4. Composites in anomalies of the TRENDY $F_{TA}$ (black lines), gross primary
productivity (GPP, green lines), terrestrial ecosystem respiration (TER, brown lines),
and the carbon flux caused by disturbances (D, blue lines) during two types of El
Niños over the extratropical northern hemisphere (NH, 23°N–90°N) and the tropics
and extratropical southern hemisphere (Trop+SH, 60°S–23°S). Shaded area denotes
the 95% confidence intervals of the variables in the composite, derived in 1000
bootstrap estimates. Bold Lines indicates the significance above the 80% level
estimated by Student's *t*-test.





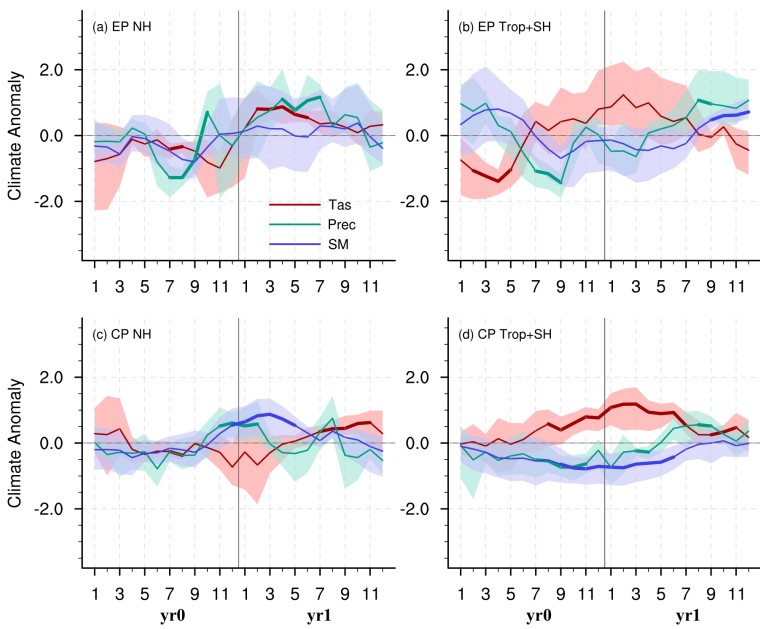


Figure 5. Composites of the standardized land surface air temperature (Tas, red lines),
precipitation (green lines), and TRENDY simulated soil moisture content (SM, blue
lines) anomalies in two types of El Niños over the NH, Trop+SH. Shaded area
denotes the 95% confidence intervals of the variables in the composite, derived in
1000 bootstrap estimates. Bold Lines indicate the significance above the 80% level
estimated by Student's *t*-test.





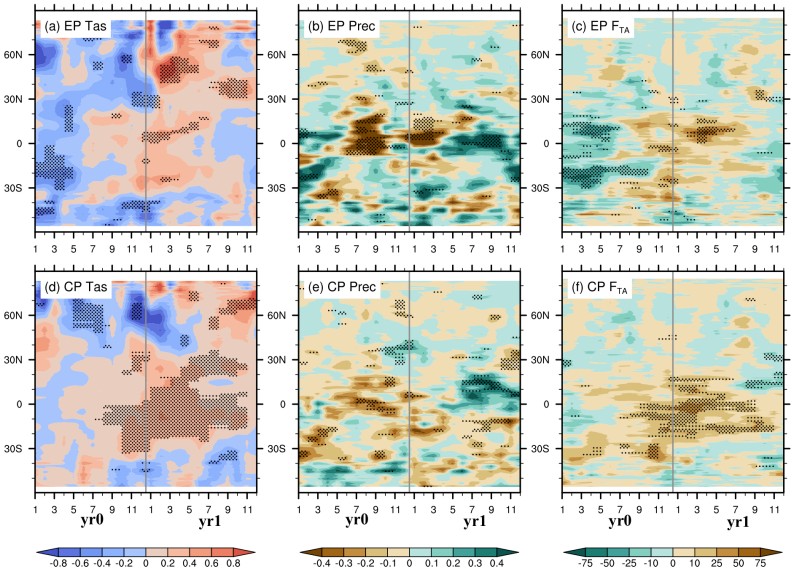


Figure 6. Hovmöller diagrams of the anomalies of climate variables and $F_{TA}$

(averaged from 180°W to 180°E) during EP and CP El Niño events. (a and d) surface

air temperature anomalies over land (units: K); (b and e) precipitation anomalies over

land (units: mm d$^{-1}$); (c and f) TRENDY simulated $F_{TA}$ anomalies (units: g C m$^{-2}$ yr$^{-1}$)

during EP and CP El Niño events, respectively. Dotted areas indicate the significance

above the 80% level estimated by Student's $t$-test.

723

724