# Peer review of "Contrasting behaviors of the atmospheric CO₂ interannual variability during two types of El Niños"

_Atmospheric Chemistry and Physics, 2018_

## Referee Comment (RC1) · Anonymous Referee #1 · 19 Apr 2018

Wang et al describe the different behaviour of CO$_2$ fluxes during the two types of El Nino event, the eastern Pacific (EP) and central pacific (CP) El Ninos. They use the atmospheric CO$_2$ growth rate and dynamic global vegetation models, and show differences for the two types of El Nino in the global CO$_2$ fluxes, as well as CO$_2$ fluxes separated regionally and by process. This is a relevant subject within the scope of ACP, the results will be useful and the paper is generally clearly written. I recommend the paper for publication after minor revision.

Detailed comments

Given the strong similarity of broad focus of this work with the recent Chylek et al paper, it might be worth adding a paragraph to the discussion that summarises the differences and similarities in approach and results e.g. exclusion of events that coincide with

volcanic eruptions, identification of different events, inclusion of TRENDY and inversion results, focus on lag by Chylek, conclusions etc. Do you also see a difference in the lag? Is there anything from the TRENDY results that could shed light on the hypothesis from Chylek that the shorter time lag between the temperature rise and an increase in CO2 emissions with CP El Ninos is influenced by fire response, while the longer time lag in EP El Ninos is dominated by vegetation response, noting although that the TRENDY models exclude or underestimate the effect of fire (maybe therefore there isn't anything you can add here, but at least worth thinking about)? Although there is a strong overlap of focus of this work with Chylek there are also significant differences, so I do believe that there is value in both studies.

Consider adding a figure (perhaps in the Supplement) with the $CO_2$ flux behaviour of separate El Nino events for EP and CP shown in comparison with the composite, to show how much the individual events vary from the composite.

page 2, line 36 - mention near the beginning of the sentence that you are considering the two types, e.g. "... evolutions of MLO CGR anomaly during the two El Nino types have three clear ..." otherwise it isn't clear until you get to the end of the long sentence.

page 2, line 44 - the sentence that begins "Regionally, significant anomalous ..." is long and you don't know which type of El Nino event this sentence refers to until the end. I suggest beginning the sentence something like "Regional analysis shows that during EP El Nino events significant anomalous ..." or some other way to mention EP at the start.

Page 5, line 111 - word "carefully" should be unnecessary

Page 7, line 154 - did the more recent version of LPX-Bern satisfy the minimum performance requirement?

Page 8, line 181 - say (broadly) what quantities you are calculating the anomalies in (e.g in model results, observations)

Page 9, line 198 - ".. with noticeable increases *in $CO_2$ growth rate* during ..."

page 9, line 210-212 - "..  and a similar regression analysis as done with the MLO CGR shows a sensitivity of 0.64 PgC $yr^{-1}$ $K^{-1}$" - Rather than describing it in this way, it would be clearer to say exactly what this is "and regression analysis of FTA with Nino3.4 shows a sensitivity of 0.64 PgC $yr^{-1}$ $K^{-1}$".

page 12, line 267 - how are you defining the MLO CGR peak here?

page 14, line 305 - "GPP anomalously increases ...etc" Can you check this sentence reflects the variations in Fig 4b? Would it be more accurate to say that there is a peak in GPP during austral fall (yr0), and is low from austral spring and winter (yr1)? Because austral summer spans from one year into the next, be more precise when you mention austral summer. Also be careful with the word increase (could be interpreted as talking about the trend) versus high values through this section.

page 16, line 349 - perhaps swap the order of figs S3 and S4 in the supplement, as S4 is always discussed before S3.

page 16, line 356-357 - "GPP is the dominant factor to FTA anomaly here" - I can see from Fig 4b that the GPP dominates globally at this time. Both GPP and TER look strongly anomalous in Feb-Aug, equator to 20N in Figs S3a and b, but the area of strongest flux is smaller for TER presumably therefore causing the dominance of GPP globally. If this is correct, maybe it is worth pointing out.

page 16, line 364 - "others" - other what? periods? regions? both?

page 17, line 378 - could mention the lag estimates from Chylek for CP and EP here.

page 18, line 402 - is there a better way to refer to this report? The url in the text did not work for me, as the new line added characters (403) to the hyperlink that shouldn't be in the url. Maybe use UNDP (2017) in the text, and remove the hyperlink from the url in the references.

[Figure]

Fig 1 - the light red shaded area is difficult to see unless the size of the figure is increased on the screen - perhaps increase the size of the figure on the page. Other figures are also small in the printed copy and it is difficult to see some of their details.

Fig 1 or text - it should be known by most people, but it wouldn't hurt to include somewhere that high values of Nino3.4 correspond to El Nino (perhaps in the Fig 1 caption or on page 6 at line 140).

Minor editing is need to improve the English in some places.
* * *

---

## Referee Comment (RC2) · Anonymous Referee #2 · 15 May 2018

This paper investigates the relationship between atmospheric CO2 inter-annual variability and El Nino events through dynamic vegetation models using the composite analysis technique. Several meteorological factors are considered in the analysis, for example, precipitation and temperature; and radiation data was not included in the analysis. The authors discussed the potential impacts radiation variability could have on the land biosphere dynamics and, subsequently, the atmospheric CO2 inter-annual variability. The title of the paper emphasizes two types of El Nino events, and the authors present a lot of details about these two types of events, but it would be great if the authors could articulate to readers why it's important to separate the two types of El Nino, and its importance to the atmospheric CO2 inter-annual variability and global carbon cycle. In general, I recommend this paper be published.

[Figure]

Some detailed comments and questions:

For the TRENDY simulations, are consistent vegetation data used amongst the models?

The composite analysis technique is very important in this study. Maybe it's better for the authors to explain briefly in the paper what this technique really is?

The English used in the paper needs further edits to eliminate some grammatical and word usage mistakes.
* * *

---

## Author Response (AR1)

**Responses to comments on "Contrasting behaviors of the atmospheric CO$_2$ interannual variability during two types of El Ninos**

Dear Referee and Editor, Thank you very much for your efforts to deal with our manuscript and provide constructive comments. We have tried our best to re-summarize the results, and modify this manuscript accordingly. The following is our point-by-point reply to the comments.

**Responses to Referee #1**

Wang et al describe the different behaviour of CO$_2$ fluxes during the two types of El Nino event, the eastern Pacific (EP) and central pacific (CP) El Ninos. They use the atmospheric CO$_2$ growth rate and dynamic global vegetation models, and show differences for the two types of El Nino in the global CO$_2$ fluxes, as well as CO$_2$ fluxes separated regionally and by process. This is a relevant subject within the scope of ACP, the results will be useful and the paper is generally clearly written. I recommend the paper for publication after minor revision.

**Detailed comments**

(1) Given the strong similarity of broad focus of this work with the recent Chylek et al paper, it might be worth adding a paragraph to the discussion that summarises the differences and similarities in approach and results e.g. exclusion of events that coincide with volcanic eruptions, identification of different events, inclusion of TRENDY and inversion results, focus on lag by Chylek, conclusions etc. Do you also see a difference in the lag? Is there anything from the TRENDY results that could shed light on the hypothesis from Chylek that the shorter time lag between the temperature rise and an increase in CO2 emissions with CP El Ninos is influenced by fire response, while the longer time lag in EP El Ninos is dominated by vegetation response, noting although that the TRENDY models exclude or underestimate the effect of fire (maybe therefore there isn't anything you can add here, but at least worth thinking about)? Although there is a strong overlap of focus of this work with Chylek there are also significant differences, so I do believe that there is value in both studies.

Reply: Thanks very much. We have added a paragraph in the discussion section to simply illustrate the differences and similarities between our work and Chylek et al. (2018). Details can be referred to the text "*As above mentioned, when finalizing our paper, we noted the publication of Chylek et al. (2018) who also focused on atmospheric $CO_2$ interannual variability during EP and CP El Niño events. We here simply illustrated some differences and similarities. In the method of the identification of EP and CP El Niño events, Chylek et al. (2018) took the Niño1+2 index and Niño4 index to categorize El Niño events, while we adopted the results of Yu et al. (2012), based on the consensus of three different identification methods, and additionally excluded the events that coincided with volcanic eruptions. The different methods made some differences in the identification of EP and CP El Niño events…*".

We can still hardly determine whether the fire response can explain the early CGR anomaly response in CP El Nino, because of TRENDY models exclude or underestimate the effect of wildfires. However, as shown in Figure 4d, the evolution of GPP anomaly in CP El Nino plays an important role in $F_{TA}$ anomaly.

Consider adding a figure (perhaps in the Supplement) with the CO2 flux behaviour of separate El Nino events for EP and CP shown in comparison with the composite, to show how much the individual events vary from the composite.

Reply: Thanks very much. We have added a figure with the CGR anomalies in the individual EP and CP El Nino events in the supplementary file (Fig. S5).

(2) page 2, line 36 - mention near the beginning of the sentence that you are considering the two types, e.g. "... evolutions of MLO CGR anomaly during the two El Nino types have three clear ..." otherwise it isn't clear until you get to the end of the long sentence.

Reply: Thanks for your constructive suggestion. We have modified it accordingly.

(3) page 2, line 44 - the sentence that begins "Regionally, significant anomalous ..." is long and you don't know which type of El Nino event this sentence refers to until the end. I suggest beginning the sentence something like "Regional analysis shows that during EP El Nino events significant anomalous ..." or some other way to mention EP at the start.

Reply: Thanks for your suggestions. We have modified it accordingly.

(4) Page 5, line 111 - word "carefully" should be unnecessary

Reply: Thanks very much. We have deleted it.

(5) Page 7, line 154 - did the more recent version of LPX-Bern satisfy the minimum performance requirement?

Reply: Thanks very much. The recent version of LPX-Bern can satisfy the requirement.

(6) Page 8, line 181 - say (broadly) what quantities you are calculating the anomalies in (e.g in model results, observations)

Reply: Thanks very much. We have modified it accordingly.

(7) Page 9, line 198 - ".. with noticeable increases *in CO2 growth rate* during ..."

Reply: Thanks very much. We have modified it as "…with noticeable increases in CGR during El Nino and decreases during La Nina, respectively".

(8) page 9, line 210-212 - ".. and a similar regression analysis as done with the MLO
CGR shows a sensitivity of 0.64 PgC yr−1 K−1" - Rather than describing it in this
way, it would be clearer to say exactly what this is "and regression analysis of
FTA with Nino3.4 shows a sensitivity of 0.64 PgC yr−1 K−1".

Reply: Thanks very much for your suggestion. We have modified it accordingly.

(9) page 12, line 267 - how are you defining the MLO CGR peak here?

Reply: Thanks very much. We have added the definition in the text. We define the
peak duration as the period above the 75% of the maximum CGR or $F_{TA}$ anomaly, in
which the variabilities of less than 3 months below the threshold are also included.

(10)page 14, line 305 - "GPP anomalously increases ...etc" Can you check this
sentence reflects the variations in Fig 4b? Would it be more accurate to say that
there is a peak in GPP during austral fall (yr0), and is low from austral spring and
winter (yr1)? Because austral summer spans from one year into the next, be more
precise when you mention austral summer. Also be careful with the word increase
(could be interpreted as talking about the trend) versus high values through this
section.

Reply: Thanks very much for your suggestions. We have checked it and modified into
"*GPP showed an anomalous positive value during austral fall (yr0), and an*
*anomalous negative value from austral fall (yr1) to winter (yr1), with the minimum*
*around April (yr1) during the EP El Niño (Fig. 4b), ...*"

(11)page 16, line 349 - perhaps swap the order of figs S3 and S4 in the supplement, as
S4 is always discussed before S3.

Reply: Thanks for your suggestion. We have swapped their order.

(12) page 16, line 356-357 - "GPP is the dominant factor to FTA anomaly here" - I can see from Fig 4b that the GPP dominates globally at this time. Both GPP and TER look strongly anomalous in Feb-Aug, equator to 20N in Figs S3a and b, but the area of strongest flux is smaller for TER presumably therefore causing the dominance of GPP globally. If this is correct, maybe it is worth pointing out.

Reply: Thanks for your suggestions. We have pointed out this and modified as "*Both GPP and TER showed the anomalous decreases (Supplementary Figs. S3a and b), and stronger decrease in GPP than in TER makes the anomalous carbon releases here (Fig. 6c).*"

(13) page 16, line 364 - "others" - other what? periods? regions? both?

Reply: Thanks. The "others" here refer to the other regions and periods. We have modified it as "*... and other regions and periods were dominated by GPP*"

(14) page 17, line 378 - could mention the lag estimates from Chylek for CP and EP here.

Reply: Thanks very much. We have mentioned the lag estimates from Chylek in the added discussion paragraph.

(15) page 18, line 402 - is there a better way to refer to this report? The url in the text did not work for me, as the new line added characters (403) to the hyperlink that shouldn't be in the url. Maybe use UNDP (2017) in the text, and remove the hyperlink from the url in the references.

Reply: Thanks very much. We have modified it as a citation "*Thomalla, F., and Boyland, M.: Enhancing resilience to extreme climate events: Lessons from the 2015-2016 El Niño event in Asia and the Pacific. UNESCAP, Bangkok.*"

(16) Fig 1 - the light red shaded area is difficult to see unless the size of the figure is
increased on the screen - perhaps increase the size of the figure on the page. Other
figures are also small in the printed copy and it is difficult to see some of their
details.

Reply: Thanks very much. We have the vectorgraph in pdf/ps format, and will supply
them to the editor during the publishing procedure.

(17) Fig 1 or text - it should be known by most people, but it wouldn't hurt to include
some- where that high values of Nino3.4 correspond to El Nino (perhaps in the
Fig 1 caption or on page 6 at line 140).

Reply: Thanks very much. Actually, in Fig. 1b we have plotted some bars in yellow
and blue which represented the CP and EP El Ninos. Correspondingly, we can see
their Nino3.4 Index in Fig.1a.

(18) Minor editing is need to improve the English in some places.

Reply: Thanks very much. We have polished the English writing by LetPub.

                    **Responses to Referee #2**

This paper investigates the relationship between atmospheric $CO_2$ inter-annual
variability and El Nino events through dynamic vegetation models using the
composite analysis technique. Several meteorological factors are considered in the
analysis, for example, precipitation and temperature; and radiation data was not
included in the analysis. The authors discussed the potential impacts radiation
variability could have on the land biosphere dynamics and, subsequently, the
atmospheric $CO_2$ inter-annual variability. The title of the paper emphasizes two types
of El Nino events, and the authors present a lot of details about these two types of events, but it would be great if the authors could articulate to readers why it's important to separate the two types of El Nino, and its importance to the atmospheric CO2 inter-annual variability and global carbon cycle. In general, I recommend this paper be published.

**Some detailed comments and questions:**

(1) For the TRENDY simulations, are consistent vegetation data used amongst the models?

Reply: Thanks for your comments. In the text, we have illustrated that TRENDY models were forced by a common set of climatic datasets (CRNCEPv6), atmospheric CO2 concentration, and land use datasets and followed the same experimental protocol. And these models are basically Dynamical Global Vegetation Models, so they do not explicitly need the vegetation data (like LAI etc.).

(2) The composite analysis technique is very important in this study. Maybe it's better for the authors to explain briefly in the paper what this technique really is?

Reply: Thanks for your suggestions. We have added a sentence to illustrate the composite analysis as "*More specifically, in terms of the composite analysis, we calculated the averages of the carbon flux anomaly (CGR, $F_{TA}$ i.e.) during the selected EP and CP El Niño events, respectively.*"

(3) The English used in the paper needs further edits to eliminate some grammatical and word usage mistakes.

Reply: Thanks for your suggestions. We have polished the English writing by LetPub.

[revised manuscript text omitted]

4a), whereas they only showed some increase during boreal summer (yr1) during the

CP El Niño (Fig. 4c).

These evolutionary characteristics of GPP, TER, and the resultant $F_{TA}$ principally resulted from their responses to the climate variability. Figure 5 shows the standardized observed surface air temperature, precipitation, and TRENDY simulated soil moisture contents. Over the Trop+SH, taking into consideration the regulation of thermodynamics and hydrological cycle on surface energy balance, variations in temperature and precipitation (soil moisture) were always opposite during the two types of El Niños (Figs. 5b and d). Additionally, adjustments in soil moisture lagged precipitation by approximately 2–4 months, owing to the so-called 'soil memory' of water recharge (Qian et al., 2008). The variations in GPP in both the El Niño types were closely associated with variations in soil moisture, namely water availability largely dominated by precipitation (Figs. 4b and 4d and 5b and 5d), and this result was consistent with previous studies (Zeng et al., 2005; Zhang et al., 2016). Warm temperatures during El Niño episodes can enhance the ecosystem respiration, but dry conditions can reduce it. These cancellations from warm and dry conditions made the amplitude of TER variation smaller than that of GPP (Figs. 4b and 4d). Over the NH, variations in temperature and precipitation were basically in the same direction (Figs.

5a and 5c), as opposed to their behaviors over the Trop+SH. This was due to the


[revised manuscript text omitted]

---

## Editor Decision (ED1)

Dear Jun Wang,

Thank-you for the work that you and your co-authors have undertaken to address the reviewers comments on your manuscript. Overall I think you have responded sufficiently to the comments from the reviewers. Below, please find a list of technical corrections that I think would improve the clarity of the manuscript in a few places. The line numbers are taken from the author response file with track-changes.

Abstract, line 207-212: Perhaps re-write this sentence to replace 'and' with 'or' i.e. 'The composite analysis showed that evolution of the MLO CGR anomaly during EP and CP El Niños had three clear differences: (1) negative or neutral precursors in the boreal spring during an El Niño-developing year (denoted as "yr0"), (2) strong or weak amplitudes, and (3) durations of the peak from December (yr0) to April during an EP El Niño-decaying year (denoted as "yr1") compared to October (yr0) to January (yr1) for a CP El Nino.

Reviewer 2 asked you to 'articulate to readers why it's important to separate the two types of El Nino' (review paragraph 1). Perhaps this could be done at line 335 by extending the sentence to note why you want to reveal their different impacts e.g. 'In this study, we attempt to reveal their different impacts given the different regional responses of the EP and CP El Ninos.'

Line 420: Suggest 'of a more recent, better performing, version'

Line 422: Suggest adding a sentence after 'experimental protocol.' 'Models use different vegetation datasets or internally generated vegetation.'

Line 445: Is the ONI plotted in Fig 1a or the Nino3.4 index without the running mean? If it is the ONI then you might like to refer to Fig 1a at line 445-446.

Line 456: After 'used here' add '(both modelled and observed)' if this is correct.

Line 571: It might be good to refer to your new supplementary figure S5 here when you are first describing the choice of years that make up each composite as well as later in the paper. You will need to re-number the supplementary figures if you do this.

Please feel free to contact me by email if any of these comments are not clear.
Regards,
Rachel Law, rachel.law@csiro.au